# The FKBP51 Inhibitor SAFit2 Restores the Pain-Relieving C16 Dihydroceramide after Nerve Injury

**DOI:** 10.3390/ijms232214274

**Published:** 2022-11-17

**Authors:** Saskia Wedel, Lisa Hahnefeld, Mohamad Wessam Alnouri, Stefan Offermanns, Felix Hausch, Gerd Geisslinger, Marco Sisignano

**Affiliations:** 1Institute of Clinical Pharmacology, Pharmazentrum Frankfurt/ZAFES, University Hospital, Goethe-University, 60590 Frankfurt am Main, Germany; 2Fraunhofer Institute for Translational Medicine and Pharmacology ITMP, Fraunhofer Cluster of Excellence for Immune Mediated Diseases CIMD, 60596 Frankfurt am Main, Germany; 3Max Planck Institute for Heart and Lung Research, 61231 Bad Nauheim, Germany; 4Center for Molecular Medicine, Goethe-University Frankfurt, Theodor-Stern-Kai 7, 60590 Frankfurt am Main, Germany; 5Department of Biochemistry, Technical University Darmstadt, 64287 Darmstadt, Germany

**Keywords:** neuropathic pain, lipid mediators, ceramides, sensory neurons, nerve injury, FKBP51

## Abstract

Neuropathic pain is a pathological pain state with a broad symptom scope that affects patients after nerve injuries, but it can also arise after infections or exposure to toxic substances. Current treatment possibilities are still limited because of the low efficacy and severe adverse effects of available therapeutics, highlighting an emerging need for novel analgesics and for a detailed understanding of the pathophysiological alterations in the onset and maintenance of neuropathic pain. Here, we show that the novel and highly specific FKBP51 inhibitor SAFit2 restores lipid signaling and metabolism in nervous tissue after nerve injury. More specifically, we identify that SAFit2 restores the levels of the C16 dihydroceramide, which significantly reduces the sensitization of the pain-mediating TRPV1 channel and subsequently the secretion of the pro-inflammatory neuropeptide CGRP in primary sensory neurons. Furthermore, we show that the C16 dihydroceramide is capable of reducing acute thermal hypersensitivity in a capsaicin mouse model. In conclusion, we report for the first time the C16 dihydroceramide as a novel and crucial lipid mediator in the context of neuropathic pain as it has analgesic properties, contributing to the pain-relieving properties of SAFit2.

## 1. Introduction

Chronic pain and especially neuropathic pain as persistent pain states affect patients worldwide after nerve injuries, chemotherapies, or diseases such as diabetes and multiple sclerosis. However, the available therapeutics for efficient pain relief are currently inadequate, pointing out neuropathic pain as a pathological entity with a lack of treatment possibilities and a broad symptom scope [1,2]. Former studies revealed FK506 binding protein 51 (FKBP51), which is encoded by the FKBP5 gene, as a novel and promising target for various pathologies, since it is involved in many different pathological processes, such as the establishment of chronic pain [3,4], stress endocrinology difficulties [5], and glucocorticoid signaling-related diseases [6]. Moreover, the highly potent FKBP51 inhibitor SAFit2 has been shown to pass the blood–brain barrier [7,8], indicating that SAFit2 is also able to pass the blood spinal cord barrier and can mediate analgesic effects in the central and peripheral nervous system. Likewise, we recently showed that SAFit2 ameliorates neuropathic pain in vivo as it counteracts the enhanced neuronal activity and reduces neuroinflammation after nerve injury [9]. These previous reports highlight the concept of FKBP51 inhibitors as a potential novel treatment approach for neuropathic pain.

Mechanistically, SAFit2 reduces the NF-κB pathway activation after nerve injury, potentially also modulating crucial target genes in the NF-κB downstream signaling [9]. NF-κB itself is a crucial transcription factor in lipid metabolism as it regulates the expression of various enzymes, such as cyclooxygenases, lipoxygenases, and phospholipases [10,11]. These enzymes metabolize and oxidize eicosanoids, linoleic acid metabolites, and sphingolipids, generating lipid mediators that can act as autocrine and paracrine signaling mediators [12]. In addition, previous studies have already demonstrated that lipid mediators can contribute to pain hypersensitivities after chemotherapeutic treatment or nerve injury [13,14,15]. Importantly, it was shown that lipid mediators can affect the sensitivity or activation threshold of ion channels in sensory neurons, especially of pain-mediating transient receptor potential cation (TRP) channels, which can result in an enhanced neuronal activity, leading to increased pain perception and neuroinflammation [14,16,17,18,19,20]. Moreover, alterations in the subset of lipid mediators were shown to be associated with nerve dysfunctions and chronic pain.

In the development of neuropathic pain, metabolites from the ceramide/sphingolipid pathway were identified as potential targets for treating neuroinflammation, as they were implicated to be crucial regulators at the neuroimmune interface [21]. Particularly, the sphingolipid sphingosine1-phosphate (S1P) is known to sensitize TRP subfamily V member 1 (TRPV1) in inflammatory processes, leading to hypersensitivity and enhanced neuropathic pain [21,22,23]. However, the influence of other S1P pathway metabolites, such as ceramides and dihydroceramides, on TRP channels has not yet been investigated in the context of neuropathic pain.

In the present study, we focus on the peripheral nervous system and the interface between the peripheral and the central nervous system. We investigated the expression of lipid-generating enzymes and performed lipidomic analysis in sensory neurons of the dorsal root ganglia, as they are the central location of the cell somata of peripheral sensory neurons, and in the dorsal spinal cord. Here, we detected that the neuropathic pain-ameliorating FKBP51 inhibitor SAFit2 also affects the expression of lipid-oxidizing enzymes, such as lipoxygenases, as well as ceramide synthases in the context of nerve injury-induced neuropathic pain in mice. An unbiased LC-HRMS-based lipidomics screening from lumbar dorsal root ganglia (DRGs) and spinal cords revealed that SAFit2 is capable of counteracting the nerve injury-induced alterations in lipid levels after SNI. While the levels of classical eicosanoids were not significantly altered, SAFit2 especially restores the levels of the C16 dihydroceramide (N-palmitoyl-D-erythro-sphinganine) after nerve injury. Neurobiological characterization of the C16 dihydroceramide showed that it significantly reduces the sensitization of TRPV1 in sensory neurons and subsequently the release of the pro-inflammatory neuropeptide CGRP (calcitonin gene-related peptide). Furthermore, the C16 dihydroceramide was shown to significantly decrease the thermal hypersensitivity of mice in an acute pain model, attributing C16 dihydroceramide potential analgesic or antihyperalgesic properties. In summary, we discovered that interfering with the FKBP51 signaling with the inhibitor SAFit2 can influence lipid signaling and metabolism in nervous tissue after nerve injury. In addition, we identified the C16 dihydroceramide as a novel and previously unrelated crucial lipid mediator in the context of neuropathic pain.

## 2. Results

### 2.1. SAFit2 Reduces the Upregulation of Lipid-Oxidizing and -Metabolizing Enzymes after SNI

To investigate the influence of SAFit2 on lipid metabolism after nerve injuries, we first analyzed the expression of lipid-oxidizing enzymes, which play a crucial role in the generation of signaling lipids and lipid metabolites, and which can influence the onset and maintenance of inflammation [12]. Therefore, we performed a spared nerve injury model on mice and treated them intraperitoneally with either 10 mg/kg SAFit2 or vehicle two times daily on six consecutive days (days five to ten) after SNI (Figure 1A). After 21 days, the mice were sacrificed, and lumbar L4-L6 dorsal root ganglia (DRGs) as well as the corresponding section of the spinal cord (SC) were collected to measure the expression of ceramide synthases, lipoxygenases, cyclooxygenases, phospholipases, and cytochrome-P_450_-oxygenases (Figure 1B–I).

Interestingly, we detected a significant reduction in ceramide synthase 5 (CerS5), which is required for the synthesis of long-chain ceramides [24], in the lumbar DRGs after SNI (Figure 1B), after conducting a one-way ANOVA with Tukey’s post-hoc test. In addition, we observed a significant difference in the expression of the arachidonate lipoxygenase 5 (ALOX5) in DRGs between the vehicle and SAFit2-treated animals (Figure 1D). However, the cyclooxygenase 2 (COX2, Figure 1F), which is typically upregulated in inflammatory processes synthesizing prostaglandins [25], shows an unaltered expression after SNI at our investigated time point. Likewise, the expression of other enzymes, such as ceramide synthase 6 (CerS6, Figure 1C), arachidonate lipoxygenase 12 (ALOX12, Figure 1E), phospholipases cPLA2 and iPLA2 (Figure 1G,H), and cytochrome-P_450_-oxygenase CYP2J6 (Figure 1I), were not significantly altered in the lumbar DRGs after SNI, when comparing treatments using one-way ANOVA with a Tukey’s post-hoc test. In addition, the expression of the arachidonate lipoxygenase 15 (ALOX15) and the cytochrome-P_450_-oxygenase CYP3a11 was below the detection limit in lumbar DRGs.

As we analyzed the expression of the same enzymes in the respective segments of the spinal cord and determined significant alterations with a one-way ANOVA and Tukey’s post-hoc test (Figure 2), we again detected an alteration in the expression of ceramide synthases and lipoxygenases. However, this time ceramide synthases 5 and 6 were both significantly upregulated in the vehicle-treated SNI animals (Figure 2B,C). In addition, the expression of the ceramide synthase 6 differs significantly between the vehicle and SAFit2-treated SNI animals, whereas it is quite comparable between SAFit2 and the sham treatment in the spinal cord after SNI (Figure 2C). Furthermore, the expression of arachidonate lipoxygenase 12 (ALOX12) was significantly altered between the vehicle and SAFit2-treated animals in the spinal cord (Figure 2E). The expression of the other lipoxygenases, cyclooxygenases, phospholipases, and cytochrome-P_450_-oxygenases was not changed in the spinal cord after SNI (Figure 2D,F–I). Expression of arachidonate lipoxygenase 15 (ALOX15) and cytochrome-P_450_-oxygenase CYP3a11 was not detectable in the spinal cord.

### 2.2. The Influence of SAFit2 on Lipid Levels after SNI

Next, we performed an untargeted lipid screening using LC-HRMS to investigate the influence of SAFit2 on lipids after nerve injury. Therefore, we analyzed the lipid distribution in L4-L6 DRGs and the respective parts of the spinal cord 21 days after SNI surgery (Figure 3). In the lipid screening, we detected 249 lipids from 14 classes—fatty acids, ceramides, diglyerides, lysophosphatidylcholines, -ethanolamines, -glycerols, -inositols, phosphatidylcholines, -ethanols, -glycerols, -inositols, -serines, sphingomyelins, and triglycerides—which can be partially further divided into subclasses (Figure 3A). For further analysis, the analytes were sorted into their lipid classes and ipsilateral values and normalized to the contralateral values, performed for every lipid. In addition, the differences between the normalized treatments were compared in heat maps for the spinal cord (Figure 3B) and lumbar DRGs (Figure 3C), separately. For lipids in the spinal cord, only slight changes were observed after nerve injury (Figure 3B). Nevertheless, some lipid classes show minor trends, such as the downregulation of fatty acids or the upregulation of lysophosphatidylethanols and phosphatidylcholines after SNI. However, these effects could be counteracted with a 10 mg/kg SAFit2 treatment after SNI. In contrast, the alterations between the treatments were stronger in the lumbar DRGs comparing the vehicle and SAFit2 treatment after SNI (Figure 3C). The classes of ceramides, diglycerides, fatty acids, lysophosphatidylglycerols, and phosphatidylcholins were clearly downregulated in the vehicle-treated SNI animals compared to the sham-treated animals. Whereas, the SAFit2 treatment counteracted this downregulation after nerve injury, adjusting the concentrations to levels comparable to the sham. In contrast, the lysophosphatidylcholins and lysophosphatidylethanols were upregulated after SNI in the vehicle-treated group while these lipids were less upregulated in the SAFit2-treated group. In summary, SAFit2 seems to compensate the SNI-induced alterations and shifts in lipid synthesis and metabolism in the respective tissues.

For investigating the effect of SAFit2 on individual lipids, we plotted the logarithmic *p*-value, which was calculated with of a two-way ANOVA and Tukey’s post-hoc test, of the two treatments against the logarithmic fold change in volcano plots for DRGs, as we have seen major differences in this tissue (Figure 3D). In Figure 3D (top), we compared the vehicle-treated SNI animals with sham animals to elucidate the impact of nerve injury on lipids, discovering four significantly altered lipids. In this comparison, the C16 dihydroceramide (d18:0/16:0), the 36:4 diglyceride, and the polyunsaturated fatty acid docosahexaenoic acid were significantly downregulated, whereas the 16:1 ether-lysophosphatidylcholine was significantly upregulated after nerve injury. Interestingly, the comparison of the vehicle and SAFit2-treated SNI animals revealed only the C16 dihydroceramide (d18:0/16:0) as a significantly altered lipid (Figure 3D (bottom)).

For showing the whole LC-HRMS dataset, the relative amount of each individual lipid is displayed in violin plots in Appendix A for DRGs, and for the spinal cord samples, in Appendix A. The internal standards and the LC-gradient used for the analysis are highlighted in Appendix A. Summarizing the results from this broad screen, we detected that some lipid mediators, such as the C16 dihydroceramide (d18:0/16:0), 36:4 diglyceride, and docosahexaenoic acid, were significantly downregulated, and the 16:1 ether-lysophosphatidylcholine was significantly upregulated after SNI. However, the C16 dihydroceramide (d18:0/16:0) was the only lipid that was significantly altered comparing the vehicle and SAFit2 treatment after nerve injury. Based on this, we hypothesized that this C16 dihydroceramide might play a crucial role in the analgesic mechanism of SAFit2.

### 2.3. C16 Dihydroceramide Desensitizes the TRPV1 Channel in Sensory Neurons

Based on the previous findings that SAFit2 reduces mechanical hypersensitivity after nerve injury [9], we investigated the influence of the C16 dihydroceramide (d18:0/16:0), which is named as C16 or C16 dihydroceramide in the following, on pain-sensing calcium channels. Firstly, we characterized the impact of C16 on the TRPV1 channel, as it is the most important and most abundantly expressed pain-mediating, ligand-gated calcium channel in sensory neurons, and plays a crucial role in the development of various pain conditions, including neuropathic pain [26].

To investigate the impact of C16 on TRPV1, we isolated sensory neurons from DRGs of naïve mice and analyzed the calcium transients of TRPV1 after C16 treatment in live cell calcium imaging (Figure 4). For comparing the calcium transients with and without C16 treatment, a protocol was established in which the TRPV1 was stimulated twice: first, only with the agonist capsaicin, and second, with the pre-incubation of C16 (2 min), to verify whether C16 has an influence on the channel-mediated calcium influx (Figure 4A). Interestingly, we detected that C16 concentration-dependently reduced the calcium transients of TRPV1 in sensory neurons (Figure 4B), after comparing the different conditions using a one-way ANOVA with a Tukey’s multiple-comparison test.

Next, we further evaluated whether this reduced calcium flux has an impact on the TRPV1 downstream signaling pathway. Therefore, the release of the peptide calcitonin gene-related peptide (CGRP) was assessed in primary sensory neuron cultures via ELISA. The neuropeptide CGRP was released from sensory neurons due to increasing intracellular calcium concentrations and functions as a mediator in neuropathic pain (Figure 4C) [27,28]. Interestingly, we observed that the C16 concentration dependently also reduced the CGRP amount in the supernatant of the primary sensory neurons, which were treated with capsaicin beforehand (Figure 4D). Furthermore, the concentration dependency was considered as statistically significant after performing a one-way ANOVA with a Tukey’s multiple-comparison test. Based on these results, we concluded that C16 reduces TRPV1-mediated calcium transients and subsequently reduces CGRP-mediated neuroinflammation.

Since many direct TRPV1 antagonists lead to severe side effects and failed in clinical trials [29], we next determined whether C16 inhibits or desensitizes the TRPV1 channel. Therefore, we changed the calcium imaging protocol in which we left out the pre-incubation step with C16, this time applying C16 together with capsaicin at the second stimulation time point (Figure 4E). However, we detected no alterations in the TRPV1-mediated calcium flux between the first and second stimulus (Figure 4F), summarizing that C16 does not directly inhibit the TRPV1 channel.

Next, we tried to elucidate how C16 reduces TRPV1-mediated calcium transients. We hypothesized that the reduced TRPV1 open probability is due to the desensitization of TRPV1, which may be indirectly mediated by a G protein coupled receptor (GPCR). To test this hypothesis, we performed a GPCR screen using three reporter assays to investigate the effect of C16 on 314 different GPCRs (Appendix A, Appendix A). However, we did not detect any C16-induced activation of a GPCR, concluding that C16 probably affects the TRPV1 open probability indirectly, possibly via interacting with the cell membrane.

### 2.4. C16 Dihydroceramide Does Not Affect TRPA1 and P2X3 in Primary Sensory Neurons

For further characterizing C16 in the context of pain, we examined the influence of C16 on other TRP channels. As a first approach, we applied 1 µM and 10 µM C16 to sensory neurons to investigate whether C16 activates any calcium channels in sensory neurons (Appendix A). However, we did not detect any calcium transients after the stimulation with C16.

Since we detected an influence of C16 on the TRPV1 channel, and since the TRPA1 channel (transient receptor potential ankyrin 1) is both co-expressed in a subgroup of TRPV1-positive sensory neurons as well as closely associated to the TRPV1, we also assessed the impact of C16 on the TRPA1 channel [30]. To examine this, we again performed a protocol involving two stimuli, however this time using allyl isothiocyanate (AITC) as the gold standard agonist of the TRPA1 channel (Figure 5A). We detected a slight increase in calcium transients after a pre-incubation with C16 compared to the vehicle control (Figure 5B). Nevertheless, the calcium transients were also increased after a pre-incubation with cremophor, using cremophor EL as a second, more lipophilic vehicle control. Based on these results, we suggest that C16 and cremophor EL does not sensitize the TRPA1; however, it increases the solubility of the lipophilic agonist AITC in Ringer’s solution, which leads to a more efficient stimulation of sensory neurons.

Moreover, we analyzed the effect of C16 on the purinergic ligand-gated calcium channel P2X3, since it also plays a critical role in the development of neuropathic pain [31]. To stimulate the P2X3 channel, we used α,β-methyleneadenosine 5′triphosphate trisodium salt (meATP) as agonist (Figure 5C). As we compared calcium transients of P2X3 with and without C16 pre-incubation, we detected no differences (Figure 5D). In summary, we observed that C16 does not directly activate any calcium channel, has a desensitizing effect on TRPV1, and no influence on the activity of TRPA1 and P2X3.

### 2.5. C16 Dihydroceramide Reduces Thermal Hypersensitivity after Capsaicin Treatment

As final step, we wanted to examine whether C16 can influence the thermal hypersensitivity of capsaicin-treated mice, as we detected a reduced TRPV1 activity after C16 treatment in vitro. Therefore, the thermal pain threshold of every mouse was assessed with the radiant heat test two times before treatment. Then, the mice were treated intraplanar with either 50 µM C16 or vehicle and 400 µM capsaicin. Afterwards, the thermal sensitivity of the mice paws was assessed over 4 h with a radiant heat test. Interestingly, we detected that the C16 treatment prevented the development of thermal hypersensitivity after capsaicin treatment in mice (Figure 6). In addition, the paw withdrawal latencies were significantly different comparing the vehicle and C16-treated mice for the time points 0.5, 1, 2, 3, and 4 h, as per assessment with a two-way ANOVA with Bonferroni’s post-hoc test. These results reveal the analgesic properties of C16, pointing out that the desensitization of TRPV1 reduces the thermal hypersensitivity of mice.

## 3. Discussion

Neuropathic pain affects around 10 percent of the general population and is induced by lesions and diseases of the somatosensory system, infections, or toxic substances [2]. However, the available therapies are very scarce because of their low efficacy and severe adverse effects [32]. Based on these circumstances, there is a high medical need for novel, safe, effective treatment approaches for neuropathic pain [32]. Nevertheless, the development of many therapy strategies failed, having faced the challenge that neuropathic pain is a multifunctional event mediated by various mediators with pleiotropic effects. In addition, the underlying mechanisms of this complex pain state are barely understood yet. In line with this, investigation of the endogenous mechanisms and alterations, for example, after nerve injury, is becoming increasingly important. Therefore, we tried to discover the alterations in lipids after SNI and assessed the influence of the potential therapeutic approach SAFit2 on these alterations.

The distribution and abundance of lipids affects various mechanisms and conditions, such as the structure of the myelin sheath, the fluidity of membranes, energy storage, and signaling pathways. In the last decades, a growing number of endogenous lipid mediators has been described to be involved in several signaling pathways and to modulate the activity of the TRP channels in sensory neurons [33]. Likewise, these endogenous lipid mediators play a crucial role in the context of persistent pain states, such as neuropathic pain and chronic pain [12,19,33,34]. Especially alterations and dysregulations in specific lipids were associated with nerve dysfunctions and neuropathic pain. A prominent example for lipid metabolism dysregulation, which results in neuropathic pain, is treatment with the cytostatic bortezomib, which significantly alters the sphingosine/ceramide pathway [20]. In contrast, the FDA-approved, S1P receptor-inhibiting substance fingolimod was shown to reduce neuropathic pain after nerve injury and in multiples sclerosis as it targets the sphingosine/ceramide pathway [35,36]. Nevertheless, alterations in the anandamide metabolism were previously associated with the development of chronic pain [37]. Furthermore, several studies have pointed out the importance of anti-inflammatory and analgesic lipid mediators, such as resolvins, in the resolution of inflammation and pain [38,39,40,41].

Here, we reveal that the selective FKBP51 inhibitor SAFit2 is capable to counteract pathophysiological alterations in lipid metabolism after SNI. More specifically, we observed that SAFit2 counteracts alterations in the expression of lipid-oxidizing enzymes as well as dysregulations in lipid classes. Likewise, the lipid classes of free fatty acids and ceramides were markedly downregulated in lumbar dorsal root ganglia and spinal cord after SNI, which were not downregulated in SAFit2-treated SNI animals. Interestingly, we could not observe any difference in arachidonic acid levels after SAFit2 treatment, indicating that the synthesis of eicosanoids, such as prostanoids, EETs (epoxyeicosatrienoic acids), and HETEs (hydroxyl-eicosatetraenoic acids), is unaffected.

Nevertheless, we detected a significant reduction in the free fatty acid (and precursor of several specialized pro-resolving mediators (SPMs)) docosahexaenoic acid (FA 22:6, DHA) in the vehicle-treated SNI mice. However, a SAFit2 treatment was capable to restore the physiological level of DHA after SNI to a level comparable to sham mice. Since docosahexaenoic acid metabolites were identified as endogenous TRP inhibitors, which can contribute to the relief of inflammation and pain [33], these results indicate that SAFit2 can restore omega-3 polyunsaturated fatty acids as precursors for potentially antihyperalgesic lipids, thereby resolving mechanical hypersensitivity after SNI [9]. Nonetheless, synthesis, signaling, and occurrence of SPM in tissues have been the subject of controversies, because of the inability and inconsistency of their detection with analytical methods in biological samples [42].

Interestingly, we detected one lipid in the untargeted screening that shows both a significant downregulation after SNI and a significantly increased level after SAFit2 treatment. These alterations reveal the C16 dihydroceramide as a novel and crucial lipid mediator in the context of neuropathic pain. However, the knowledge about ceramides is very limited in the research field of pain, although the group of sphingosines from the sphingosine/ceramide pathway are well characterized in the context of neuropathic pain [21,22,23,43,44,45,46,47,48]. Based on that, we decided to assess the impact of the C16 dihydroceramide on pain mediating TRP channels and its function in acute pain. We showed that the C16 dihydroceramide reduces the sensitivity of the mechanical and thermal pain-mediating TRPV1 channel. However, it does not inhibit the calcium channel directly, which is quite a crucial property, since many TRPV1 antagonists lead to hyperthermia as they disturbed the complex regulatory role of TRPV1 for normal body temperature maintenance [13,29,49]. Based on that, it might be beneficial to maintain the TRPV1 activity in a physiological state and to reduce its sensitization, which mainly occurs in inflammation-related pain states. In conclusion, the desensitizing effects of the C16 dihydroceramide further contribute to the analgesic effects of SAFit2, which are capable of mediating pain relief after neve injury [9].

It is still unclear how SAFit2 alters the levels of the C16 dihydroceramide. Possibly, SAFit2 is involved in regulating the expression of ceramide synthase 5 (CerS5) in sensory neurons. This may be caused by the interference of SAFit2 in NF-κB signaling [9]. In addition, it has previously been shown that NF-κB signaling can be influenced by the ceramide synthase 4 [50]. SAFit2 may also enhance or restore activity or expression of phospholipases, leading to enhanced release and availability of free fatty acids after treatment.

Since we detected no direct effect of the C16 dihydroceramide on TRPV1, we analyzed whether the lipid mediator might desensitize the TRPV1 indirectly via a G protein coupled receptor. However, we also did not detect any activation of a G protein coupled receptor, when screening a library consisting of 314 GPCRs. Based on these results, we suggest that the C16 dihydroceramide modulates the TRPV1 indirectly, as it alters the biophysical plasma membrane properties or may act as an allosteric modulator of TRPV1, as was previously described for cholesterol and other lysophospholipids [51,52,53]. In addition, we observed an influence of the C16 dihydroceramide on the downstream signaling of TRPV1, as it reduces the release of the pro-inflammatory neuropeptide CGRP in sensory neurons. CGRP is secreted upon enhanced neuronal activity and can cause neuroinflammation via GPCRs, leading to the development of persistent pain states [27,28,54]. We further confirmed these analgesic properties in an acute mouse model in which we could show that the C16 dihydroceramide significantly reduced capsaicin-induced thermal hypersensitivity to a physiological normal level. Here, we identify the C16 dihydroceramide as a novel antihyperalgesic lipid mediator and observe the effects of SAFit2 on the expression of ceramide synthase 5 and on the free fatty acid levels in nervous tissue. These findings help explain the mechanism of SAFit2-meditated analgesic effects and they provide additional evidence for the involvement of lipids as a regulatory interface for antihyperalgesia in nervous tissue in the context of neuropathic pain.

A limitation of the study is the lack of female mice in our in vivo experiments. Previous studies revealed evidence for sex differences in inflammatory pain models and the development of neuropathic pain [55]. However, the SAFit2-inhibiting target FKBP51 was previously shown to be sex-independently involved in the development of neuropathic pain [4]. In addition, the sex dimorphism in inflammatory and neuropathic pain models was attributed to the involvement of different immune cells in the development of pain [56]. However, our in vivo study focused on acute TRPV1-mediated nociception, which is sex independent and does not involve the influence of any immune cells.

In summary, we report that the highly selective FKBP51 inhibitor SAFit2 is capable of counteracting the alterations in lipid metabolism and signaling after SNI. More specifically, SAFit2 restored the levels of the C16 dihydroceramide, which reduced the TRPV1 activity, as well as the subsequent secretion of CGRP in primary sensory neurons. Both effects indicate the analgesic properties of the C16 dihydroceramide, revealing this lipid as a novel and crucial lipid mediator in the context of neuropathic pain.

Collectively, these findings can mechanistically explain the analgesic effects of SAFit2 in vivo. They also raise the question whether C16 dihydroceramide itself can serve as a treatment for neuropathic pain. However, due to the short half-life and limited distribution and bioavailability of this lipid [57,58], we do not consider it as a direct therapeutic option for the treatment of neuropathic pain.

## 4. Materials and Methods

### 4.1. Spared Nerve Injury Surgery and SAFit2 Treatment

For all animal experiments, we obtained wild type mice with a C57BL/NRj background from commercial breeding companies, such as Janvier and Charles River. Mice were matched regarding age (8–10 weeks) for the in vivo treatments. For analyzing the effect of SAFit2 after nerve injury, we conducted the well-characterized and robust spared nerve injury (SNI) model. Therefore, surgery was performed under anesthesia, in which we firstly ligated the common peroneal and tibial branches of the sciatic nerve and then dissected the sciatic nerve distally from the ligature on the level of the knee joint. The third nerve branch was left intact [59]. After surgery, we treated the mice intraperitoneally with either 10 mg/kg SAFit2 or vehicle (PBS supplemented with 5% Tween, 5% PEG400, and 0.7% ethanol) twice daily from Day 5 to 10.

### 4.2. Tissue Isolation

On Day 21 after surgery, the mice were sacrificed, and lumbar dorsal root ganglia (DRGs) (L4–L6) and the respective segments of the spinal cord (SC) were dissected. The respective tissue was isolated from injured (ipsilateral) and unimpaired (contralateral) sites and frozen in liquid nitrogen for either RNA isolation or LC-HRMS analysis.

### 4.3. Quantitative Real-Time PCR

The total RNA was isolated from lumbar DRGs (L4-L6) and the respective segments of spinal cord using the mirVana miRNA Isolation Kit (Applied Biosystems, Waltham, MA, USA). The isolation was performed according to the manufacturer’s instructions. As a next step, we measured the RNA concentrations using a NanoDrop ND-1000 spectrophotometer (NanoDrop Technologies, Wilmington, DE, USA). The reverse transcription was conducted with 200 ng RNA for DRG samples and 400 ng RNA for spinal cord samples using the First Strand cDNA Synthesis Kit (Thermo Fisher Scientific, Waltham, MA, USA), compliant with the manufacturer’s instructions. Quantification of gene expression was assessed with a quantitative real-time PCR, which was performed using the TaqMan^®^ Gene Expression Assay System with respective assay primers (Thermo Fisher Scientific) and the QuantStudio™ Design & Analysis Software v 1.4.3 (Thermo Fisher Scientific). The experimental settings were adjusted compliant to the manufacturer’s instructions and using the assay primers listed in Table 1. The raw data were evaluated with the ΔΔC(T) method to calculate the relative expression of the target genes, as described previously [60,61].

### 4.4. Liquid Chromatography–High Resolution Mass Spectrometry (LC-HRMS)

For LC-HRMS purposes, DRGs and spinal cord (SC) were homogenized using a pellet pestle mixer (Thermo Fisher scientific). A volume of 79 or 237 µL of ethanol:water (1:9, *v*/*v*) was added to approximately 1 mg (DRGs) or 3 mg (SC) of tissue and grinded for 30 to 60 s, including short breaks to avoid overheating.

For lipid extraction, 75 µL of internal standards, solved in methanol (see Appendix A), were added to 20 µL of tissue homogenate. Next, 250 µL of methyl-tert-butyl-ether (MTBE, HPLC-grade) and 50 µL of 50 mM ammonium formate (for mass spectrometry, ≥99.0 was purchased from Sigma-Aldrich, Munich, Germany) were added and the samples mixed for 1 min. After 5 min of centrifugation at ambient temperature and 20,000× *g*, the upper phase was transferred and the lower phase reextracted using 100 µL of MTBE:methanol:water (10:3:2.5, *v*/*v*/*v*, upper phase), again followed by centrifugation. The combined upper phases were dried under a nitrogen stream at 45 °C, stored at −80 °C and redissolved in 100 µL of methanol prior to analysis. For analysis, an Exploris 480 Orbitrap mass spectrometer coupled with a Vanquish horizon LC-system (both Thermo Fisher Scientific) was used.

Chromatographic separation was performed using a Zorbax RRHD Eclipse Plus C8 column (1.8 µm 50 × 2.1 mm ID, Agilent) with a precolumn of the same type and a 14-min binary gradient. The samples were analyzed in positive and negative ionization mode with a scan range from 180 to 1500 m/z and 120,000 mass resolution. Additionally, data-dependent spectra were acquired at a 15,000 resolution and a total cycle time of 600 ms. Acquisition was performed using XCalibur software v4.4, and for evaluation, TraceFinder software v5.1 was applied (both Thermo Fisher Scientific). Lipids were identified as previously described, with a mass error of 5 ppm [62]. The first ten spinal cord samples were pooled, and multiple replicates were analyzed over the run-time to verify system stability. The results were normalized to the protein concentration, which was determined after tissue homogenization with a Bradford assay.

### 4.5. Isolation of Dorsal Root Ganglia (DRGs)

For performing assays with primary sensory neurons, naïve mice were sacrificed and DRGs isolated, which were stored in precooled HBSS, supplemented with CaCl_2_ and MgCl_2_ (Gibco), on ice during dissection until further purification. For purification purposes, we substituted HBSS with a collagenase/dispase solution containing 500 U/mL collagenase and 2.5 U/mL dispase diluted in neurobasal medium (Gibco). The isolated DRGs were incubated with the collagenase/dispase solution for 75 min at 37 °C. Next, we removed the enzyme solution by centrifuging and discarding the supernatant, followed by two washing steps with neurobasal medium, supplemented with 10% FCS (*v*/*v*). Afterwards, the DRGs were further enzymatically dissociated via an incubation step with 0.05% trypsin (*v*/*v*) (Gibco) for 10 min at 37 °C. The DRGs were again washed twice with neurobasal medium containing 10% FCS. After the enzymatic dissociation, we dissociated the DRGs mechanically in neurobasal medium (Gibco) supplemented with L-glutamine (2 mM; Gibco), penicillin (100 U/mL; Gibco), streptomycin (100 µg/mL; Gibco), B-27 supplement (Gibco), and gentamicin (50 µg/mL; Gibco). Finally, 30 µL of the primary sensory neuron suspension were plated on poly-L-lysine-coated cover slips or 48-well plates and incubated at 37 °C for two hours to allow adhesion of the sensory neurons. After the attachment phase, we added another 2 mL of neurobasal medium to each dish.

### 4.6. Calcium Imaging

To analyze the calcium levels in primary sensory neurons, we stained sensory neuron cultures with Fura-2-AM for at least 45 min prior the measurement. Afterwards, the cells were washed twice with freshly prepared Ringer’s solution. We also used Ringer’s solution during live cell calcium imaging as we measured the baseline levels. In addition, we applied Ringer’s solution after and in between stimulations for wash-out purposes. The Ringer’s solution consisted of 145 mM NaCl, 1.25 mM CaCl_2_ × 2 H_2_O, 1 mM MgCl_2_ × 6 H_2_O, 5 mM KCl, 10 mM D-glucose, and 10 mM HEPES, and its pH was adjusted to a physiological level of 7.3 using NaOH. To analyze the influence of the C16 dihydroceramide on different calcium channels, we pre-incubated primary sensory neurons with the C16 dihydroceramide for two minutes (0.25–5 µM) and stimulated the cells with the following agonists, respectively: 100 nM capsaicin for 30 s (TRPV1 agonist), 100 µM allyl isothiocyanate for 45 s (AITC, TRPA1 agonist), and 50 µM α,β-methyleneadenosine 5′triphosphate trisodium salt (meATP, P2X3 agonist). Control experiments were performed with the respective volume of the corresponding vehicle, which was either DMSO or cremophor EL. The agonists as well as the C16 dihydroceramide were diluted from stock solutions in Ringer’s solution to their final concentrations. The calcium live cell imaging measurements were conducted using an automated perfusion system (ValveBank II, AutoMate Scientific, San Diego, CA, USA) and recorded with a DMI4000 B Microscope, with a compact light source CTR550 HS (Leica Microsystems, Wetzlar, Germany).

### 4.7. Calcitonin Gene-Related Peptide Assay

To investigate the influence of the C16 dihydroceramide on calcitonin gene-related peptide (CGRP) secretion, the amount of CGRP was measured in the supernatant of sensory neurons after TRPV1 stimulation. Therefore, sensory neurons were isolated from DRGs as previously described and stimulated with a mixture of capsaicin and C16 dihydroceramide the next day. To prepare the primary sensory neurons for the secretion assay, we washed them with HBSS previously. Afterwards, we treated the cells with either vehicle as negative control, capsaicin as positive control, or a mixture of capsaicin and C16 dihydroceramide for 15 min. All compounds were diluted in HBSS. The amount of CGRP in the supernatant was detected with an ELISA kit from Bertin Bioreagent, in accordance with the manufacturer’s instructions, measuring the absorption at 405 nm. The quantification was performed with linear regression, as described in the manufacturer’s manual.

### 4.8. Screen for C16 Dihydroceramide Activated GPCRs

#### 4.8.1. β-Arrestin Assay

In total, 5000 HTLA cells were seeded into a white, transparent, and ploy-L-Lysin-coated 384-well plate from PerkinElmer. The cells were co-transfected after 6 h with a plasmid from the PRESTO-Tango GPCR library (Addgene, Watertown, MA, USA, [63]). We used a mixture of 10 ng plasmid and 0.04 µL Lipofectamine 2000 per well when performing the transfection, as described by [63]. We used GFP as a transfection control and 100 µM carbachol and the muscarinic M5 receptor as a positive control. After 24 h, the medium was replaced by 45 µL of serum-free medium. The ligand (5 µL) was then added at a final concentration of 30 µM for approximately 24 h. Afterwards, the medium was aspirated, and the cells lysed using 50 µL of bright-Glo reagent (Promega, Madison, WI, USA) diluted 10 times with PBS. After 15 min of incubation with lysis buffer, the luminescence (endpoint, 1500 ms integration time) was measured using a flexstation 3 plate reader.

#### 4.8.2. cAMP Reporter Assay

In total, 5000 HEK293 cells were seeded in a white, transparent and ploy-L-Lysin-coated 384-well plate from PerkinElmer. The cells were co-transfected after 6 h with the cAMP reporter pGL4.29 plasmid (Promega) and a plasmid from the PRESTO-Tango GPCR library (10 ng from each plasmid and 0.04 µL Lipofectamine 2000 per well). As a transfection control, we used GFP and as a positive control we used 10 nM GLP-1 acting on the GLP-1 receptor. After 48 h, the medium was replaced by 45 µL of serum-free medium for one hour. The ligand (5 µL) was then added at a final concentration of 15 µM for 5 h. Afterwards, the medium was aspirated, and the cells were lysed using 50 µL of a mixture of bright-Glo reagent (Promega) diluted 10 times with PBS. After 15 min of incubation with lysis buffer, the luminescence (endpoint, 1500 ms integration time) was determined using a flexstation 3 plate reader.

#### 4.8.3. RhoA Reporter Assay

In total, 5000 HEK293 cells were seeded in a white, transparent and ploy-L-Lysin coated 384-well plate from PerkinElmer. The cells were co-transfected after 6 h with the pGL4.34 plasmid (Promega) and a plasmid from the PRESTO-Tango GPCR library (10 ng from each plasmid and 0.04 µL Lipofectamine 2000 per well). As a transfection control, we used GFP and as a positive control we used FBS on cells transfected only with the reporter plasmid pGL4.34. The medium was aspirated after 24 h, and serum-free medium was added instead. After 48 h, the medium was replaced by 45 µL of serum-free medium, and cells were incubated for one more hour. The ligand (5 µL) was then added at a final concentration of 15 µM for 5 h. Afterwards, the medium was aspirated, and the cells were lysed using 50 µL of a mixture of bright-Glo reagent (Promega) diluted 10 times with PBS. After 15 min of incubation with lysis buffer, the luminescence (endpoint, 1500 ms integration time) was measured using a flexstation 3 plate reader.

### 4.9. Radiant Heat Assay

During the behavioral experiment, the experimenter was blinded. All animals were transferred into respective test cages for at least one hour before the measurement to allow habituation, as described previously [64]. For assessing the thermal withdrawal latency, a radiant heat test was performed using a Plantar Analgesia Meter with a high-intensity projector lamp (IITC Life Science, Woodland Hill, CA, USA). The cut-off time for this experiment was set to 20 s.

### 4.10. Data Analysis and Statistics

All animal numbers and in vitro replicates are depicted in the figure legends. In addition, the data are displayed as the mean ± SEM. The statistical analysis was performed with GraphPad Prism software version 9. The normal distribution of the raw data was verified and confirmed with the Shapiro–Wilk test. We performed an unpaired and heteroskedastic Student’s *t*-test with Welch’s correction for comparing two groups for the in vitro experiments. To determine the significance between more than two groups, we conducted a one-way analysis of variance (ANOVA). In line with this, we used a two-way ANOVA when comparing more than three groups. To evaluate the significance levels for behavioral assays, we performed a two-way ANOVA with Bonferroni’s multiple-comparisons post-hoc test. We considered a *p*-value lower than 0.05 as statistically significant.

## Figures and Tables

**Figure 1 ijms-23-14274-f001:**
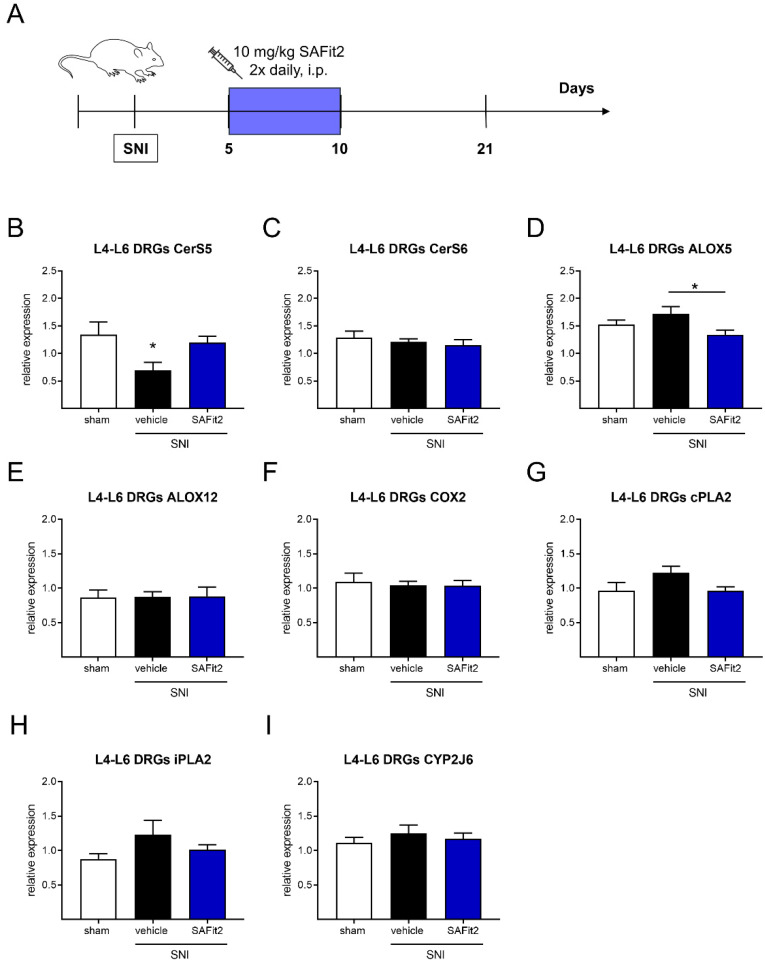
Gene expression of ceramide synthases, lipoxygenases, phospholipases, and epoxygenases in lumbar DRGs of nerve-injured mice after additional SAFit2 treatment. (**A**) Schematic illustration of the experimental procedure: SNI surgery was performed followed by a treatment with either 10 mg/kg SAFit2 or vehicle on six consecutive days (days five to ten) after SNI. The gene expression of ceramide synthases (**B**,**C**), lipoxygenases (**D**–**F**), phospholipases (**G**,**H**), and epoxygenase (**I**) was measured 21 days after the surgery. The data are displayed as the mean ± SEM of technical replicates from three to four mice per treatment. * *p* < 0.05, as per one-way ANOVA with Tukey’s post-hoc test. Abbreviations: DRGs: dorsal root ganglia; SAFit2: selective antagonist of FKBP51 by induced fit 2; SNI: spared nerve injury; CerS: ceramide synthase; ALOX: arachidonate lipoxygenase; COX2: cyclooxygenase 2; cPLA2: cytosolic phospholipase A2; iPLA2: calcium-independent phospholipase A2; CYP2J6: cytochrome P450 family 2 subfamily j polypeptide 6.

**Figure 2 ijms-23-14274-f002:**
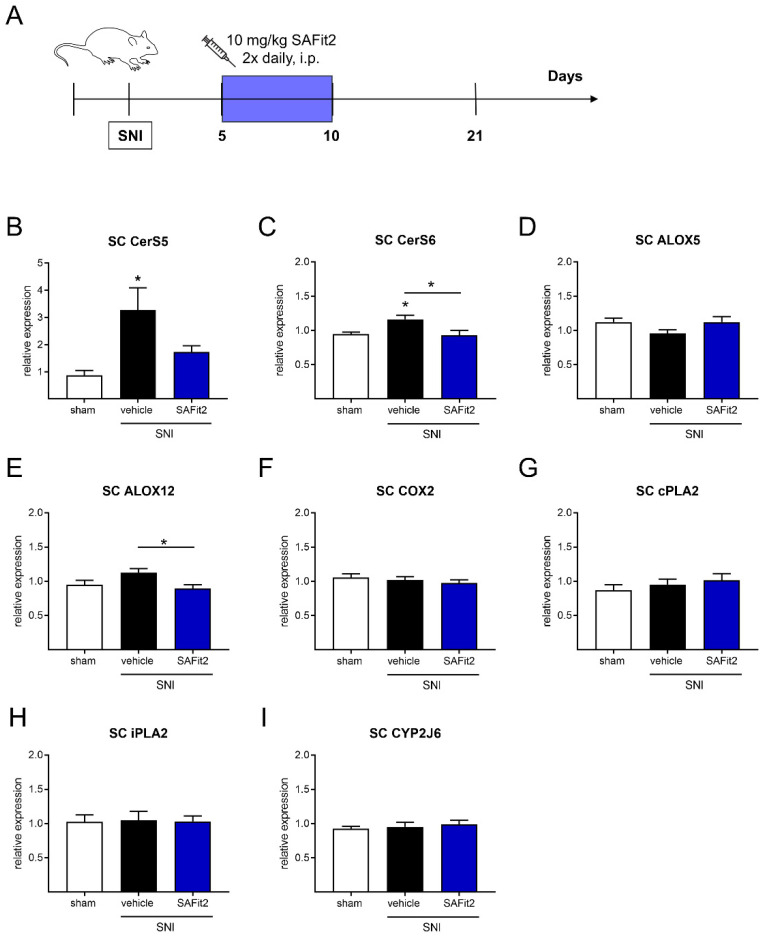
Gene expression of ceramide synthases, lipoxygenases, phospholipases, and epoxygenases in the spinal cord of nerve-injured mice after additional SAFit2 treatment. (**A**) Schematic illustration of the experimental procedure: SNI surgery was performed followed by treatment with either 10 mg/kg SAFit2 or the vehicle on six consecutive days (days five to ten) after SNI, as depicted in Figure 1. The gene expression of ceramide synthases (**B**,**C**), lipoxygenases (**D**–**F**), phospholipases (**G**,**H**), and epoxygenase (**I**) was measured 21 days after the surgery. The data are displayed as the mean ± SEM of technical replicates of three to four mice per treatment. * *p* < 0.05, as per one-way ANOVA with a Tukey’s post-hoc test. Abbreviations: SAFit2: selective antagonist of FKBP51 by induced fit 2; SNI: spared nerve injury; CerS: ceramide synthase; ALOX: arachidonate lipoxygenase; COX2: cyclooxygenase 2; cPLA2: cytosolic phospholipase A2; iPLA2: calcium-independent phospholipase A2; CYP2J6: cytochrome P450 family 2 subfamily j polypeptide 6.

**Figure 3 ijms-23-14274-f003:**
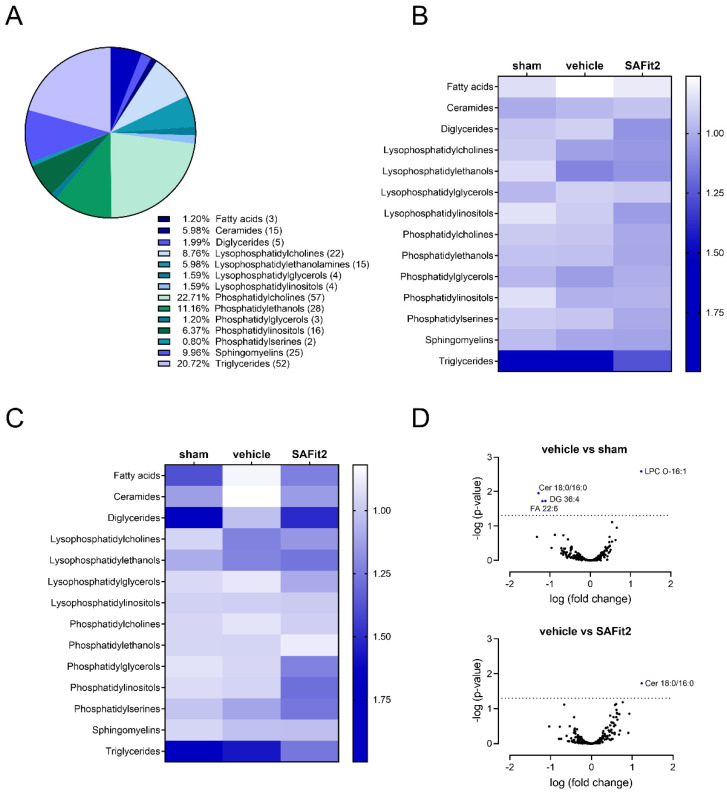
LC−HRMS lipid analysis of lumbar DRGs (L4−L6) and the respective segments of spinal cord from nerve-injured mice after additional SAFit2 treatment. (**A**) Distribution of the number of measured analytes per lipid class in LC-HRMS analysis. (**B**) Abundancy of lipid classes in the spinal cord of the vehicle and 10 mg/kg SAFit2 treated SNI animals as well as sham animals. (**C**) Lipid levels of the lipid classes in lumbar DRGs of the vehicle and 10 mg/kg SAFit2-treated animals after SNI, and the sham animals. (**D**) Volcano plots comparing the lipid levels of the sham and vehicle-treated SNI animals (top) and vehicle and SAFit2-treated animals after SNI (bottom). A *p*-value of 0.05 was determined as a threshold for significance, which was calculated with a two-way ANOVA with a Tukey’s post-hoc test. All significantly altered lipids were labeled and depicted in blue. Abbreviations: DRGs: dorsal root ganglia; LC−HRMS: liquid chromatography–high-resolution mass spectrometry; SAFit2: selective antagonist of FKBP51 by induced fit 2; SNI: spared nerve injury.

**Figure 4 ijms-23-14274-f004:**
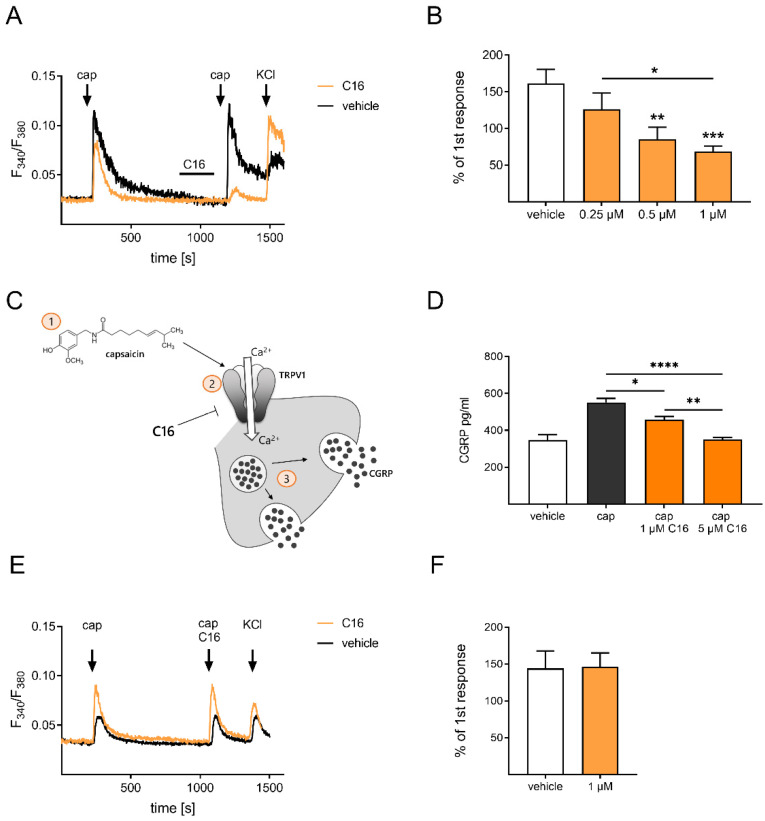
C16 dihydroceramide causes a desensitization of the TRPV1 channel and subsequently leads to a reduced CGRP release from primary sensory neurons. (**A**) Representative calcium responses of sensory neurons, which were pre-treated with C16 for 2 min, followed by an activation of the TRPV1 channel with the agonist capsaicin (100 nM, 30 s). Afterwards, the cells were treated with potassium chloride (50 mM, 45s) to discriminate reacting cells as neurons. (**B**) Quantification of the two calcium responses in (**A**) after capsaicin stimulation (untreated and C16 treated). (**C**) The schematic illustration displays the downstream signaling of TRPV1 after capsaicin activation, which results in the release of CGRP from sensory neurons. A scheme depicting the mechanism was designed using images from motifolio. (**D**) CGRP amount measured in the supernatant of C16 and capsaicin treated sensory neurons. (**E**) Representative calcium responses of sensory neurons, which were treated with both capsaicin (100 nM, 45 s) and C16 (1 µM, 45 s) at the same time point. (**F**) Quantification of the double stimulation from (**E**), ruling out antagonist properties of C16. (**B**,**D**) Each condition displays the mean ± SEM of 23–38 measured primary sensory neurons. * *p* < 0.05, ** *p* < 0.01, and *** *p* < 0.001, and **** *p* < 0.0001, as per one-way ANOVA with a Tukey’s post-hoc test. Abbreviations: CGRP: calcitonin gene-related peptide; cap: capsaicin; KCl: potassium chloride; TRPV1: transient receptor potential cation channel subfamily V member 1.

**Figure 5 ijms-23-14274-f005:**
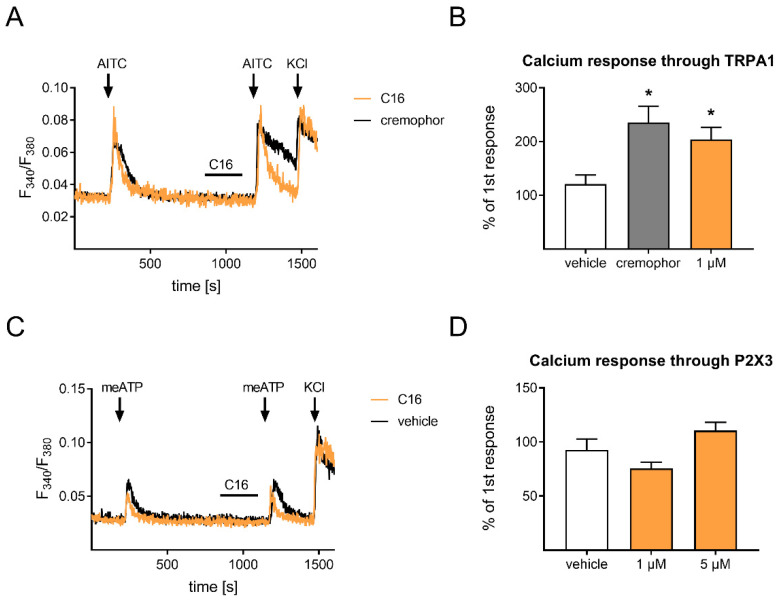
C16 dihydroceramide has no influence on the activity of TRPA1 and P2X3 in primary sensory neurons. (**A**) Representative calcium responses of sensory neurons, which were pre-incubated with C16 (2 min) and stimulated with the TRPA1 agonist AITC (100 µM, 45 s) followed by a KCl (50 mM) treatment as a positive control. (**B**) Quantification of the two calcium responses in (**A**) after AITC stimulation (untreated and C16 treated). (**C**) Representative calcium responses of sensory neurons, which were pre-incubated with C16 (2 min) and stimulated with the P2X3 agonist meATP (50 µM, 30 s) followed by a KCl (50 mM) treatment as a positive control. (**D**) Quantification of the two calcium responses in (C) after meATP stimulation (untreated and C16 treated). (**B**,**D**) The data represents the mean ± SEM of 23–30 sensory neurons per group. * *p* < 0.05, as per one-way ANOVA with a Tukey’s post-hoc test. Abbreviations: TRPA1: transient receptor potential cation channel subfamily A member 1; AITC: allyl isothiocyanate; meATP: α,β-methyleneadenosine 5′triphosphate trisodium salt; KCl: potassium chloride.

**Figure 6 ijms-23-14274-f006:**
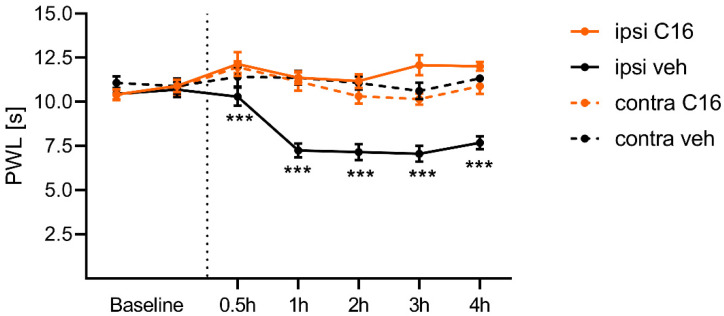
C16 dihydroceramide reduces the thermal hypersensitivity after capsaicin treatment in mice. We analyzed the baseline levels the day before and on the day of treatment. The mice were treated intraplantarly with either capsaicin and the C16 dihydroceramide or capsaicin and vehicle to analyze the analgesic properties of C16. The thermal pain threshold was assessed 30 min, 1 h, 2 h, 3 h, and 4 h after treatment. The data represent the mean ± SEM from seven mice per group., *** *p* < 0.001, as per two-way ANOVA with Bonferroni’s post-hoc test.

**Table 1 ijms-23-14274-t001:** List of used TaqMan^®^ Gene Expression Assays.

Target	Gene	Article Number	Company
ALOX12	Arachidonate 12-lipoxgenase	Mm00545833_m1	Thermo Fisher
ALOX15	Arachidonate 15-lipoxgenase	Mm00507789_m1	Thermo Fisher
ALOX5	Arachidonate 5-lipoxgenase	Mm01182747_m1	Thermo Fisher
CerS5	Ceramide synthase 5	Mm00556165_m1	Thermo Fisher
CerS6	Ceramide synthase 6	Mm00510998_m1	Thermo Fisher
COX2	Cytochrome c oxidase subunit II	Mm03294838_g1	Thermo Fisher
CYP2J6	Cytochrome P450, family 2, subfamily j, polypeptide 6	Mm01268197_m1	Thermo Fisher
CYP3a11	Cytochrome P450, family 3, subfamily a, polypeptide 11	Mm00731567_m1	Thermo Fisher
GAPDH	Glceraldehyde-3-phosphate dehydrogenase	Mm99999915_g1	Thermo Fisher
PLA2g4a	Phospholipase A2, group 4a	Mm00447040_m1	Thermo Fisher
PLA2g4c	Phospholipase A2, group 4c	Mm01195718_m1	Thermo Fisher

## Data Availability

Not applicable.

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
