# Peer review of "The FKBP51 Inhibitor SAFit2 Restores the Pain-Relieving C16 Dihydroceramide after Nerve Injury"

_ijms, 2022, doi:10.3390/ijms232214274_

Round 1

Reviewer 1 Report

Despite the broad expression of the results, I have not been able to directly observe the statistics obtained after the analyses. At some points it was difficult for me to follow the reading and I could only observe the significance obtained. This makes it difficult to be able to replicate the analyses themselves because I do not know what test they have done at each point. Curiously, it appears in some figure captions, but not in the text, which only shows the interpretation of the results. Please rewrite the material and methods section because self-plagiarism has been detected.

Author Response

Reviewer #1:
Despite the broad expression of the results, I have not been able to directly observe the statistics obtained after the analyses. At some points it was difficult for me to follow the reading and I could only observe the significance obtained. This makes it difficult to be able to replicate the analyses themselves because I do not know what test they have done at each point. Curiously, it appears in some figure captions, but not in the text, which only shows the interpretation of the results. Please rewrite the material and methods section because self-plagiarism has been detected.

To address the issue of the reviewer, we added all information concerning statistical analysis in the respective results sections of the manuscript. This information includes statistical tests and post-hoc tests that were used, as well as sample size for the experiments. The summary of all statistical evaluation that was performed in the manuscript is also depicted in section 4.10. Data analysis and statistics.
We also substantially rewrote and reformulated the materials and methods section, as well as other parts of the manuscript to avoid indications of self-plagiarism.

Reviewer 2 Report

This is a very interesting study concerning the modulation of lipid signaling by SAFit2 to alleviate the neuropathic pain. The modulation pathway was mainly through the TRPV1 channel, which involved with the levels of the C16 dihydroceramide. The study was very comprehensive, but there were some points that the authors should address clearly.

First, the neuropathic pain pathway was involved from the paw skin to the somatosensory cortex. Why the authors only show the data in dorsal root ganglion?

Did you have any data in dorsal spinal cord or somatosensory cortex? Did they did not reveal any significance?  

Second, why you sacrifice the animals at day 21 to determine the expression profile and why not at day 28? Concerning the neuropathic pain, the determination of expression profile should be as longer as possible.    

Third, in Figure 6, the data showed the alleviation of acute pain but not chronic pain, which is not compatible the modulation of neuropathic pain depicted in the beginning of the abstract. The authors should have the animal behaviors such as mechanic or thermal withdrawn latency or intensity subjected SNI determined at different time profile from the base line to 28 days.          

Author Response

Reviewer #2

This is a very interesting study concerning the modulation of lipid signaling by SAFit2 to alleviate the neuropathic pain. The modulation pathway was mainly through the TRPV1 channel, which involved with the levels of the C16 dihydroceramide. The study was very comprehensive, but there were some points that the authors should address clearly.

First, the neuropathic pain pathway was involved from the paw skin to the somatosensory cortex. Why the authors only show the data in dorsal root ganglion?

We agree with the reviewer that neuropathic pain affects all levels of the peripheral and central nervous system. However, in this study, we focused on the peripheral nervous system and the interface between the peripheral an central nervous system for our analysis because they are most directly affected after nerve injury by sensitization processes and neuroinflammation (Ji, Chamessian et al. 2016, Berta, Qadri et al. 2017, Ji, Nackley et al. 2018).

Since we wanted to investigate expression of lipid generating enzymes in sensory neurons, we used dorsal root ganglia as they are the central location of the cell somata of peripheral sensory neurons and the dorsal spinal cord as interface location between the peripheral an central nervous system. We would not be able to analyze expression changes in sensory neurons using paw tissue or somatosensory cortex.

After we found differences in the expression levels, we performed our lipid screen in the same tissues to compare lipid levels with expression changes.

To address the issue of the reviewer, we included the following text passage in the introduction section of our manuscript:

In the present study, we focus on the peripheral nervous system and the interface between the peripheral and the central nervous system. We investigated the expression of lipid generating enzymes and performed lipidomic analysis in sensory neurons of the dorsal root ganglia, as they are the central location of the cell somata of peripheral sensory neurons, and in the dorsal spinal cord.

Did you have any data in dorsal spinal cord or somatosensory cortex? Did they did not reveal any significance?  

As depicted in figures 2 and 3 in our manuscript we also analyzed expression of lipid generating and oxygenating enzymes in the dorsal spinal cord. We also performed our lipid screen in dorsal spinal cord. However, we did not see strong alterations in lipid levels after SAFit2 treatment in the spinal cord. In DRGs, we could observe stronger effects, especially concerning free fatty acids and ceramides (fig 3b vs 3c).

However, as described above, we focus on the peripheral nervous system and the interface between the peripheral and the central nervous system in this study, which is why we did not investigate the somatosensory cortex.

Second, why you sacrifice the animals at day 21 to determine the expression profile and why not at day 28? Concerning the neuropathic pain, the determination of expression profile should be as longer as possible.    

This is an important point raised by the reviewer. Usually, expression changes in DRGs and spinal cord occur within the first days to 14 d after nerve-injury (Berta, Perrin et al. 2017, Cai, Zhao et al. 2018, Ahlstrom, Matlik et al. 2021). Also, microglia activation in the spinal cord increases transiently but decreases again several months after nerve injury (Tansley, Uttam et al. 2022). According to these studies, the strongest expression changes are expected between days 0 and 14 after nerve injury, and there should not be strong expression differences between days 21 and 28 post nerve-injury.

Another reason for us to perform our expression and lipidomic experiments on days 21 post nerve-injury, is that SAFit2-tretament occurred between days 5 and 10 after nerve injury (see fig 1A in the manuscript). We chose the time point for our analyses to be close to the SAFit2-treatment period to relate the observed effects to SAFit2 treatment. At later timepoints, it would have been more difficult to relate any effects to the previous SAFit2 treatment, because treatment and analysis time points would have been too long apart.  

The combination of these considerations led us to day 21 after SNI for our analyses.

Third, in Figure 6, the data showed the alleviation of acute pain but not chronic pain, which is not compatible the modulation of neuropathic pain depicted in the beginning of the abstract. The authors should have the animal behaviors such as mechanic or thermal withdrawn latency or intensity subjected SNI determined at different time profile from the base line to 28 days.   

We thank the reviewer for this comment because we did not explain this in the manuscript in detail. We previously showed the effect of the FKBP51 inhibitor SAFit2 on nerve-injury-induced neuropathic pain (Wedel, Mathoor et al. 2022).

In the present manuscript, we try to identify the effects of SAFit2 on the lipidome in nervous tissue respectively and see that SAFit2 restores the levels of C16 ceramide in DRGs after nerve injury. The identified C16 ceramide is capable of transiently reducing thermal hypersensitivity in vivo. This could explain the analgesic effects of SAFit2 that we reported in our previous manuscript. It also highlights the use of SAFit2 as a potential therapeutic option for the treatment of neuropathic pain.

However, although C16 ceramide can reduce thermal hypersensitivity in vivo, we do not believe that the C16 ceramide itself can serve as a treatment for neuropathic pain because of its short half-life (Chen, Berejnaia et al. 2018, Zelnik, Volpert et al. 2020).

To address this issue, we included the following text passage in the discussion section of our manuscript:

Collectively, these findings can mechanistically explain the analgesic effects of SAFit2 in vivo. They also raise the question whether C16 dihydroceramide itself can serve as a treatment for neuropathic pain. However, due to the short half-life and limited distribution and bioavailability of this lipid ( reference Chen, Berejnaia et al. 2018, Zelnik, Volpert et al. 2020), we do not consider it as a direct therapeutic option for the treatment of neuropathic pain
